# The economic burden of non-communicable disease mortality in the South Pacific: Evidence from Fiji

**Shamal Shivneel Chand**[⊗], **Baljeet Singh**[iD]*[⊗], **Sanjesh Kumar**[⊗]

School of Economics, Faculty of Business and Economics, The University of the South Pacific, Suva, Fiji

⊗ These authors contributed equally to this work.
* singh_bl@usp.ac.fj

**Data Availability Statement:** All relevant data are within the manuscript and its Supporting Information files.

**Funding:** The author(s) received no specific funding for this work.

## Abstract

Non-communicable diseases (NCDs) have emerged as one of the major endemics in Fiji which is responsible for more than 80 percent of deaths annually. In this study, we estimate the economic burden of non-communicable disease mortality in Fiji. The specific impact of diabetes, cardiovascular disease, chronic respiratory disease and cancer-related mortality on Fiji's output is also investigated using the autoregressive distributed lag bounds tests approach to cointegration. The data used is compiled from Fiji Ministry of Health and Medical Services and World Health Organization's Mortality database. Overall, the study finds that NCD mortality rate together with cardiovascular disease, diabetes, chronic respiratory disease and cancer have a significant negative effect on output per capita of Fiji between 1972 and 2016. A one percentage point increase in NCD-mortality rate reduced output per capita by 0.012 percent. In addition, a percentage point increase in the mortality rates of cardiovascular disease, diabetes, chronic respiratory disease and cancer decreased output per capita by 0.018, 0.01, 0.031, and 0.035 percent, respectively. The findings conclude that NCD poses significant economic burden in Fiji and recommend policy innovations in lessening the high risk of NCD among the Fijian population.

## Introduction

The prevalence rate for non-communicable diseases (NCDs) in the low and middle-income countries has been growing steadily, presenting a major threat to people, families, and communities while hindering the potential achievement of development goals [1]. Each year, 41 million people die from an NCD of which 15 million people are between ages 30 and 69 [2]. Rising NCD crisis in the low and middle-income countries poses a significant threat to the progress on sustainable development goals especially the poverty reduction strategies. In particular, low income earners and disadvantaged group are more likely to get sick and die from NCDs because of greater risk of getting exposed to behavioural risk factors such as unhealthy diet, and harmful consumption of tobacco and alcohol, which makes households use family income to finance their healthcare cost [3]. The global economic burden of NCD study also

**Competing interests:** The authors have declared that no competing interests exist.

reported that NCDs are likely to cause around US$47 trillion in output loss within the next two decades [4].

Despite being a developing nation, Fiji has one of the highest rates of NCDs in the world where it accounts for more than 80 percent of all deaths, of which most are premature [5]. The four major types of NCDs that include cardiovascular diseases (CVD), diabetes mellitus (DM), chronic respiratory disease (CRD) and cancers (CAN) accounts for the largest share with cardiovascular diseases such as heart disease and stroke affecting most number of Fijians [6]. In 2018, Fiji recorded the highest death rate from diabetes compared to any other country with 188 fatalities per 100,000 [7]. The burden of NCD-related deaths on Fiji's output is under-researched, despite the high rate of NCD prevalence and mortality. Some of studies involving cost-of-illness analysis estimated FJ$8.8 million in cost arising from stroke mortality among young people [8] and approximately FJ$49 million in output loss from rheumatic heart disease related premature mortality annually [9]. These studies have been influential on the cost of NCDs in Fiji; however, they only provide cost estimate of specific type of NCD mortality.

The macroeconomic studies argue that NCDs are detrimental to the level of economic output and growth through the channels of labour and capital accumulation [10–12]. Overtime, NCDs will reduce the quality and quantity of a country's labour force, affecting and lowering the national income. Workers with NCDs are more likely to get regularly sick, which will reduce their efficiency in terms of using machinery and equipment in production whereas NCD mortality will reduce the size of skilled and unskilled labour essential for long run economic growth [13]. Additionally, NCD-related morbidity will reduce the capital stock since savings will be used for the treatment of NCDs instead of investment purposes [4]. While NCD could also indirectly affect the development of human capital when children are required to take care of their sick parents from young age.

NCDs, in particular, diabetes, cardiovascular disease, chronic respiratory disease and cancer are growth-retarding factors. Although the literature shows limited empirical investigation in this area, most of the cost-of-illness studies have concluded that NCDs pose a significant financial burden on the households, individuals, businesses and the whole economy [14–18]. The empirical studies that did include NCDs into the growth models have found that it significantly reduced the economic growth and long-run output. For example, Suhrcke and Urban [19] used the GMM one-step system estimation and regressed the gross domestic product (GDP) per capita on its five-year time lag, openness, average years of schooling, investment rate, fertility rate, adult mortality rate and importantly the CVD mortality rate. The study concludes that CVD is detrimental to growth only for countries with a high per capita income or at a threshold income of US$7231. Since NCDs were still lagging behind communicable diseases in developing countries in the year of study, the detrimental effects were not noticeable. However, Suhrcke and Urban [19] argued that as NCDs become more common in the developing countries because of changing lifestyles, its adverse effect on health, income and growth will become more noticeable and common, despite what is the level of per capita income in the country.

Health is a multidimensional concept that cannot be fully measured using a single indicator. Hence, one of the widely used measure for health is the probability of death, which is captured by life expectancy and the infant mortality rate [20]. Other measures of health could be the Disability Adjusted Life Years lost (DALY), which does not only include the measures of mortality but also the measures of morbidity due to an increase in disease incidence. Intelligence, such as the national average intelligence quotient (IQ) and test scores, can also be a measure of health since it falls in the biological aspect of human development and education. Improvement in biological health over time is an indication of improved intelligence and

human capital [20]. This study uses the NCD mortality rate as an indicator of health capital to measure its impact on Fiji's per capita income level.

Despite fewer studies investigating linkages between NCDs and economic growth, a few studies have looked at the relationship between nutritional status and economic output since nutritional status is an important link between NCDs and growth relationship. Using the cointegration approach and Granger causality test, Neeliah and Shankar [21] attempted to derive a short-run and long-run causality between calorie intake and GDP. The study recommends that calorie intake needs to be minimised to reduce the NCD prevalence in the population. A similar study by Dube and Phiri [22] in the context of South Africa found a positive and significant relationship between nutritional intake and economic growth. The estimated coefficient of nutritional intake was 0.15 and the authors found a strong causal effect from nutrition to economic growth.

Theoretically, NCDs will reduce the supply of labour and productivity. As the mental and physical capacity of the worker deteriorates due to NCD morbidity, the level of productivity, efficient use of technology and machinery diminishes [19]. NCD-related morbidity and sickness in the labour force imply that more workers will be out of the work-force to get their treatment while mortality will permanently reduce the size of the skilled labour force [10]. In this context, the firms endure an additional cost of training and hiring new workers for the positions left vacant by workers suffering from a particular type of NCD. Additionally, NCD-related morbidity increases the healthcare cost and decreases capital investment since individuals suffering from NCDs use savings for treatment [10]. It would be rather difficult to factor in all the channels into the model due to the lack of data on morbidity, disability and healthcare cost devoted to NCDs. Two of the studies have used NCD-related mortality rates to measure the impact of NCD on output, particularly Suhrcke and Urban [19] for OECD countries and Frank [23] for a panel of Latin America and Caribbean countries.

Hence, in the spirit of the studies mentioned, this study will also use the NCD mortality rate to measure the impact of NCD on the output of Fiji. Autoregressive Distributed Lag (ARDL) bounds test approach to cointegration is used to examine the relationship. Being one of the first studies in this area for Fiji, the findings will be of utmost importance and relevant to the health and economic sector. The rest of the paper is structured as follows. Section 2 outlines the methods and materials inclusive of model, estimation procedure and data source. The results are discussed in section 3 and lastly, section 4 provides concluding remarks and policy implications.

## Methods and materials

We employ the model developed by Bloom, Canning [24] to demonstrate the impact of the NCD mortality rate on Fiji's output. The model has physical capital, labour, human capital, and health capital and technology as exogenous components in the production function as follows.

$$Y = AK^{\propto}L^{\beta}e^{\emptyset_1 hc + \emptyset_2 h} \tag{1}$$

where $Y$ is the output, measured by real gross domestic product (RGDP); $A$ is the total factor productivity (TFP); $K$ is the stock of physical capital; and $L$ is the labour force. Bloom, Canning [24] decomposed productivity in terms of human capital ($hc$) and health status ($h$). In the model, human capital consisted of three components, namely average years of schooling, average work experience of the work force, and squared average work experience. On the other hand, life expectancy is used as a proxy for health. For the purpose of this analysis, gross secondary school enrolment rate is used as a proxy for human capital [25]. As the objective of this study postulates, mortality rates of aggregate NCD, diabetes, cardiovascular disease, chronic respiratory disease and cancer will be used as the proxy for health status component, similar to

the study by Frank [23]. According to Bloom, Canning [24], the functional form as outlined in Eq (1) has an advantage since it replicates the microeconomic studies where wage, as in output in macroeconomics studies, depends on schooling and health status of workers.

The following equations are derived after transforming the model in per capita terms and substituting the secondary school enrolment rate as human capital and NCD mortality rates into health capital.

$$\text{Major NCDs} : \ y = Ak^{\propto}e^{\theta_1 SEC + \theta_2 NCD} \tag{2}$$

$$\text{Decomposed Major NCDs} : \ y = Ak^{\propto}e^{\theta_3 SEC + \theta_4 DM + \theta_5 CVD + \theta_6 CRD + \theta_7 CAN} \tag{3}$$

where $y$ is output per capita, $k$ is physical capital per capita, and $SEC$ is the gross secondary school enrolment rate in Eqs (2) and (3). NCD mortality rate ($NCD$) is the proxy for health capital in Eq (2). In Eq (3), $DM$ is diabetes, $CVD$ is cardiovascular disease, $CRD$ is the chronic respiratory disease and $CAN$ is the cancer mortality rate. Hence, we take natural logarithms *(ln)* of Eqs (2) and (3) to derive the linear forms for estimation as follows.

$$\text{Major NCDs} : \ lny_t = a_t + \propto lnk_t + \theta_1 SEC_t + \theta_2 NCD_t + e_t \tag{4}$$

$$\begin{aligned}\text{Decomposed Major NCDs} : \ lny_t \\ = a_t + \propto lnk_t + \theta_1 SEC_t + \theta_2 DM_t + \theta_3 CVD_t + \theta_3 CRD_t + \theta_4 CAN_t + e_t\end{aligned} \tag{5}$$

The term $e$ represents the residual in the equations above. We expect both $k$ and $SEC$ to have positive impact on output per capita whereas NCD mortality rates are expected to be negatively related to output. Similar studies in this area by Frank [23] demonstrated that NCD mortality rate has a negative impact on output per capita of Latin America and Caribbean countries. In addition, Suhrcke and Urban [19] also conclude that cardiovascular disease and NCD mortality rates negatively affect the future growth of output per capita, particularly in high-income countries. Models (4) and (5) are estimated using the ARDL bounds test to cointegration technique, and more detail of this approach is outlined in the estimation procedure.

## Estimation procedure

The economic theory stipulates that there exists a long-run relationship among the macroeconomic variables when the mean and variances are constant over time and not trended. However, recent years of empirical researches have shown that time-series variables, in particular, have trended mean and variances or are simply non-stationary. Hence, ordinary least squares (OLS) estimates give spurious results that could be disastrous for policymaking.

The non-stationarity problem in the time-series variables usually arise from the presence of unit root or structural breaks, usually dealt with by de-trending or taking the first difference of the variables. Although this allows estimation of short-run dynamics, the long-run information is lost. To solve the issue of non-stationarity among the time-series variables, econometric analysis has moved towards the cointegration method. Cointegration among the variables occurs when two or more time-series variables are related in a way that in the long-run it moves to some steady state equilibrium. The variables need to be integrated of the same order I(d) while the residuals need to be integrated of order one less I(d-1) for the cointegration to occur, according to Engle and Granger (1987). If the variables are cointegrated then it is possible to estimate the long-run equation. On the other hand, if the variables do not cointegrate then the long-run estimation will give spurious regression and it will only be possible to estimate the short-run dynamics.

Granger [26] was the first to suggest the possibility of cointegration among variables in a spurious regression. Engle and Granger [27], autoregressive distributed lag bounds test approach to cointegration [28, 29], and Johansen and Juselius [30] cointegration techniques have largely contributed towards the cointegration literature. Additionally, each of the technique estimates the error correction model to derive the short-run dynamics and correction of disequilibrium in a long-run scenario.

The classical Engle and Granger [27] cointegration technique only allows for cointegration tests to be conducted when the variables are integrated of the same order. For the purpose of this study, autoregressive distributed lag bounds test approach to cointegration by Pesaran, Shin [29] is adopted as an appropriate methodology to investigate the existence of long-run level relationship and derive efficient estimates since some of the time-series variables in the investigation were initially found to be I(1) while other variables were I(0) at level forms. Hence, one of the advantages of the ARDL model is that it is very flexible where variables of both I(0) and I(1) status can be used to test for a cointegrating relationship. Secondly, the ARDL approach is an advantage to studies using a small sample size for estimation and forecasting [31]. Thirdly, the ARDL approach gives a consistent and unbiased estimate of the long-run parameters and each variable can have their own different lag-lengths compared to conventional cointegration tests [32]. The fourth advantage of the ARDL approach is that it adequately deals with the problems of autocorrelation and endogeneity and provides unbiased and super-consistent coefficients with valid t-statistics [32–34]. In sum, there are two steps involved in the ARDL bounds test approach to cointegration.

**Step 1: Investing the existence of long-run relationship.** The calculated F-statistics for bounds test to cointegration determines the existence of a long-run relationship among the variables. The bounds F-statistic is calculated when one of the variables stand as an endogenous variable while others are exogenous. According to Narayan and Smyth [35], if a long-run relationship among the variables is predicted by the ARDL methodology then the error correction regression (Step 2) can be estimated without having any knowledge on the direction of the long-run relationship among the variables. The following general ARDL regression outlines the overall cointegration procedure.

$$\Delta Y_t = \delta_0 + \sum_{i=1}^{k} \alpha_1 \Delta Y_{t-i} + \sum_{i=0}^{k} \alpha_2 \Delta X_{t-i} + \delta_1 Y_{t-1} + \delta_2 X_{t-1} + v_t \qquad (6)$$

The following equation outlines the ARDL regressions for models (4) and (5):
*Model 4 ARDL regression*

$$\Delta lny_t = \emptyset_0 + \sum_{i=1}^{k} \rho_1 \Delta lny_{t-i} + \sum_{i=0}^{k} \rho_2 \Delta lnk_{t-i} + \sum_{i=0}^{k} \rho_3 \Delta SEC_{t-i} + \sum_{i=0}^{k} \rho_4 \Delta NCD_{t-i} + \emptyset_1 lny_{t-1}$$
$$+ \emptyset_2 lnk_{t-1} + \emptyset_3 SEC_{t-1} + \emptyset_4 NCD_{t-1} + \emptyset_5 COUP_t + w_t \quad (7)$$

*Model 5 ARDL regression*

$$\Delta lny_t = \varphi_0 + \sum_{i=1}^{k} \sigma_2 \Delta lny_{t-i} + \sum_{i=0}^{k} \sigma_3 \Delta lnk_{t-i} + \sum_{i=0}^{k} \sigma_4 \Delta SEC_{t-i} + \sum_{i=0}^{k} \sigma_5 \Delta DM_{t-i}$$
$$+ \sum_{i=0}^{k} \sigma_6 \Delta CVD_{t-i} + \sum_{i=0}^{k} \sigma_7 \Delta CRD_{t-i} + \sum_{i=0}^{k} \sigma_8 \Delta CAN_{t-i} + \varphi_1 lny_{t-1} + \varphi_2 lnk_{t-1}$$
$$+ \varphi_3 SEC_{t-1} + \varphi_3 DM_{t-1} + \varphi_4 CVD_{t-1} + \varphi_5 CRD_{t-1} + \varphi_6 CAN_{t-1} + \varphi_7 COUP_t + x_t \quad (8)$$

In the above ARDL regressions, *k* is the optimum number of lags selected by the Akaike Information Criterion (AIC). The joint null hypotheses for testing the long-run relationship among the variables in Eqs (7) and (8) are stated as follows: The null hypotheses for testing the long-run relationship are $\emptyset_1 = \emptyset_2 = \emptyset_3 = \emptyset_4 = \emptyset_5 = 0$ for Eq (7) and $\varphi_1 = \varphi_2 = \varphi_3 = \varphi_4 = \varphi_5 = \varphi_6 = \varphi_7 = 0$ for Eq (8).

The upper and lower bounds critical values provided by Pesaran, Shin [29] for large sample size and Narayan [31] for small sample size are used for testing existence of a long-run relationship. If the F-statistic lies below the lower bound critical value at 5 percent significance level, the test fails to reject the null hypothesis (no existence of long-run relationship). On the other hand, if the F-statistic lies above the upper bound critical value, the test rejects the null hypothesis. However, in the case of F-statistic falling within the lower and upper bound critical values, the test for cointegration is inconclusive.

**Step 2: Estimating the Error Correction Model (ECM).** Time-series variables at level form suffer from non-stationarity problems and one of the ways to deal with non-stationarity is to first difference the variables. However, the estimated equation using the first-differenced variables give short-run parameters while the long-run information is lost. Therefore, the short-run dynamics are not useful in certain ways since the policy-makers and researchers are more interested in the long-run properties. Thus, the error correction models incorporate both the short-run and long-run dynamics.

The general ECM for the ARDL specification can be written in terms of the first difference and lagged level variables form as follows:

$$\Delta Y_t = \delta_0 + \sum_{i=1}^{k} \beta_i \Delta Y_{t-i} + \cdots + \gamma_1 ECT_{t-1} + \varepsilon_t \tag{9}$$

The $ECT_{t-1}$ is the error correction term defined as:

$$ECT_{t-1} = Y_{t-1} - \sum_{i=0}^{k} \theta_i X_{i,t-1} \tag{10}$$

The ECM specification (9) is applied to models (4) and (5):

*Model 4 ECM*

$$\Delta lny_t = \emptyset_0 + \sum_{i=1}^{k} \rho_1 \Delta lny_{t-i} + \sum_{i=0}^{k} \rho_2 \Delta lnk_{t-i} + \sum_{i=0}^{k} \rho_3 \Delta SEC_{t-i} + \sum_{i=0}^{k} \rho_4 \Delta NCD_{t-i}$$
$$+ \emptyset_1 COUP_t + \emptyset_2 ECT_{t-1} + w_t \tag{11}$$

*Model 5 ECM*

$$\Delta lny_t = \varphi_0 + \sum_{i=1}^{k} \sigma_1 \Delta lny_{t-i} + \sum_{i=0}^{k} \sigma_2 \Delta lnk_{t-i} + \sum_{i=0}^{k} \sigma_3 \Delta SEC_{t-i} + \sum_{i=0}^{k} \sigma_4 \Delta DM_{t-i}$$
$$+ \sum_{i=0}^{k} \sigma_5 \Delta CVD_{t-i} + \sum_{i=0}^{k} \sigma_6 \Delta CRD_{t-i} + \sum_{i=0}^{k} \sigma_7 \Delta CAN_{t-i} + \varphi_1 COUP_t + \varphi_2 ECT_{t-1}$$
$$+ x_t \tag{12}$$

The $ECT_{t-1}$ is derived as residuals from the cointegrating equation which shows disequilibrium correction between the previous period ($t-1$) and the current period ($t$) or the adjustment parameter. The coefficient of $ECT_{t-1}$ is expected to be negative and should lie between zero and a negative one to show convergence. For example, if the parameter is -0.5 then it shows that 50 percent of the disequilibrium is adjusted within the current period and the model converges to a steady state. A positive parameter indicates model instability and movement away from the steady-state, which has no relevance in policy-making.

## Data

Table 1 provides the list of all variables used in the analysis and their data sources. The study uses time-series macroeconomic data from 1972 to 2016. The GDP data is from the World Bank's development indicators database at constant Fiji dollars while physical capital stock is from the Penn World Tables v9.0. It was important to use real values to remove the influence of inflation for efficient estimations. Output per capita $y$ and capital per capita $k$ is calculated by dividing the aggregate GDP and capital stock with the total population.

**Table 1. Data definition, description and sources.**

| Variables | Description | Source(s) |
|---|---|---|
| *y* | Output per capita. GDP at constant 2010 FJ$ divided by the total population. | GDP at constant 2010 FJ$ and the total population was sourced from the World Bank's database. |
| *k* | Capital stock per capita. Physical capital stock in constant FJ$ divided by total population. | Data for the capital stock is sourced from the Penn World Tables v9.0 |
| *DM* | Diabetes Mellitus mortality rate (%) | Fiji Ministry of Health's Annual Reports and WHO's Mortality database |
| *CVD* | Cardiovascular disease mortality rate (%) | Fiji MOH's Annual Reports and WHO's Mortality database |
| *CRD* | Chronic Respiratory disease mortality rate (%) | Fiji MOH's Annual Reports and WHO's Mortality database |
| *CAN* | Cancer mortality rate (%) | Fiji MOH's Annual Reports and WHO's Mortality database |
| *NCD* | Non-communicable diseases mortality rate (%) | Fiji MOH's Annual Reports and WHO's Mortality database |
| *SEC* | Secondary School gross enrolment ratio (%). | World Bank's development indicators database |
| *COUP* | The dummy variable for political instabilities in years 1987, 2000 and 2006. | Author's calculation |

The main variables of interest in this study are the mortality rates for NCDs, diabetes, cardiovascular disease, chronic respiratory disease and cancer. The mortality rates were computed by dividing the number of deaths for a particular disease with the total number of deaths in a given year. Data on the number of deaths caused by NCDs were acquired from the Fiji Ministry of Health and Medical Services Annual Reports and World Health Organization's mortality database with reference to International Classification of Diseases (ICD) 10. After compiling the number of deaths for all major NCDs, it was divided by the total number of deaths in a given year (from 1972–2016) to get the mortality rates.

Table 2 provides a summary and distribution of the main variables used in the analysis from 1972 to 2016. On average, GDP per capita was FJ$6,669.63, capital per capita was FJ$21,759.41 and secondary school enrolment rate was 77.95 percent between 1972 and 2016. While average cardiovascular disease, diabetes, cancer and chronic respiratory disease mortality rates were 36.42, 9.85, 10 and 8.23 percent respectively.

## Results and discussion

### Unit root test

Prior to testing for cointegration, we test the order of integration for each variable using the Augmented Dickey-Fuller (ADF) test. The unit root test equation for ADF is depicted as

**Table 2. Summary statistics from 1972–2016.**

| Variable | Observations | Minimum | Maximum | Mean | Std. deviation |
|---|---|---|---|---|---|
| Output per capita | 45 | 5103.88 | 8836.71 | 6669.63 | 1022.04 |
| Capital per capita | 45 | 9180.40 | 40052.34 | 21759.41 | 8781.94 |
| SEC (%) | 45 | 58.85 | 91.01 | 77.95 | 8.56 |
| CVD (%) | 45 | 23.03 | 49.24 | 36.42 | 5.00 |
| DM (%) | 45 | 1.76 | 23.85 | 9.85 | 7.27 |
| CAN (%) | 45 | 5.69 | 15.04 | 10.00 | 2.05 |
| CRD (%) | 45 | 4.70 | 13.47 | 8.23 | 2.34 |
| NCD (%) | 45 | 48.93 | 75.37 | 64.50 | 7.14 |

**Table 3. Unit root test.**

| Variable | Lag length | ADF-Statistics | Critical Value | Conclusion |
|---|---|---|---|---|
| $lny$ | 0 | -2.54 | -3.52 | I(1) |
| $\Delta lny$ | 0 | -8.95 | -2.93 | I(0) |
| $lnk$ | 1 | -2.85 | -3.52 | I(1) |
| $\Delta lnk$ | 0 | -5.74 | -2.93 | I(0) |
| NCD | 7 | -2.88 | -3.54 | I(1) |
| $\Delta$NCD | 4 | -6.16 | -2.94 | I(0) |
| DM | 0 | -2.54 | -3.52 | I(1) |
| $\Delta$DM | 0 | -6.94 | -2.93 | I(0) |
| CVD | 0 | -3.33 | -3.52 | I(1) |
| $\Delta$CVD | 1 | -6.68 | -2.93 | I(0) |
| CRD | 0 | -6.66 | -3.51 | I(0) |
| $\Delta$CRD | 9 | -3.99 | -2.95 | I(0) |
| CAN | 1 | -2.32 | -3.52 | I(1) |
| $\Delta$CAN | 0 | -11.13 | -2.93 | I(0) |
| SEC | 0 | -3.40 | -3.19 | I(0) |
| $\Delta$SEC | 0 | -7.77 | -2.93 | I(0) |

The null hypothesis indicates that the series has a unit root problem. Up to 9 lags were tested and AIC was used to select the optimum number of lags. Variables at level form included both the intercept and trend, however, only intercept was included in the first difference equations. All unit root tests were conducted in EViews 10. Critical values for ADF-statistics are provided at a 5 percent significance level.

$\Delta y_t = a_0 + \theta y_{t-1} + \gamma t + a_1 \Delta y_{t-1} + a_2 \Delta y_{t-2} + \cdots + a_p \Delta y_{t-p} + a_t$ where the joint null hypotheses are $H_0$: $\theta = 0$ (non-stationary) and $H_1$: $\theta < 0$ (stationary). Although not required by the ARDL framework, we conduct the unit root tests to ensure that none of the variables are integrated of order two since that will make the provided critical values for bounds test invalid. As can be seen from Table 3, only chronic respiratory disease (CRD) and secondary school enrolment rate (SEC) are stationary at 5 percent at level form, whereas all other variables are stationary at first difference. Hence, the results indicate that the ARDL framework is the only appropriate method to analyse the long-run cointegration relationship rather than the Engle-Granger and the Johansen cointegration model.

## ARDL bounds test approach to cointegration

We use critical values provided by Narayan [31] for bounds test since the critical values provided by Pesaran, Shin [29] are based on large sample size, which is not appropriate for time-series studies involving small sample size. For model (4), the calculated F-statistic is higher than the upper bound critical value at 10 percent significance level and for model (5); the F-statistic is above the upper bound critical value at 5 percent significance level (Table 4). Hence, the null hypothesis of no cointegration is rejected and there exists a long-run cointegration relationship among the variables in both models. After establishing that a long-run cointegration relationship exists among the variables, we estimate each equation using the unrestricted error correction model. The maximum lag length was set at 4 in EViews and the optimum lag length was chosen using the Akaike Information Criterion (AIC).

## Long-run and short-run results

After establishing that a long-run cointegration relationship exists among the variables, we estimated each equation using the unrestricted error correction model (UECM). We present

**Table 4. F-statistics for ARDL bounds test for cointegration.**

| Models | Critical value bounds of the F-statistic | | | | Calculated F-Statistic |
|--------|------|------|------|------|------|
| | 5% level | | 10% level | | |
| | $I(0)$ | $I(1)$ | $I(0)$ | $I(1)$ | |
| (4) | 3.08 | 4.02 | 2.56 | 3.43 | 3.56[C] (k = 3) |
| (5) | 2.59 | 3.77 | 2.19 | 3.25 | 4.72[B] (k = 6) |

A, B, C indicate significance at 1, 5 and 10 percent levels. The F-statistic values are compared to the critical values by Narayan [31]. k is the number of regressors.

the long-run and short-run empirical results for model (4) in Tables 5 and 6 respectively. The empirical results for model (5) are presented in Tables 7 and 8. Additionally, we included the standard diagnostic tests for each model alongside the short-run results.

Prior to interpreting the long-run and short-run results, we assess the model stability and validity of parameters using the standard diagnostic and stability tests. The error correction terms ($ECT_{t-1}$) in the short-run UECM for models (4) and (5) are statistically significant at 1 percent level and have a negative sign, which confirms that a long-run cointegration relationship exists among the variables. The error correction coefficients are -0.698 for model (4) and -0.777 for model (5), which indicates that 69.8 percent and 77.7 percent of the disequilibrium in models (4) and (5) from past periods are corrected in the current period respectively. The overall goodness of fit for each model is also quite good, with the Adjusted-R squared values of 47 percent for model (4) and 68 percent for model (5). We found no evidence of serial correlation, heteroscedasticity, non-normality of errors and improper functional form in model (4) and model (5). In addition, Figs 1 and 2 show the cumulative sum of residuals (CUSUM) and cumulative sum of squared residuals (CUSUM of squares) stability test results for models (4) and (5), respectively, which are within the 5 percent critical bounds indicating both models are stable. Hence, the statistics for error correction term, diagnostic tests and model stability test confirm that the long-run and short-run coefficients are stable and indeed affect Fiji's output per capita.

In the long-run, the coefficient of capital per worker (*lnk*) has a positive sign in models (4) and (5), but the parameter values are unstable. The elasticity of capital per capita is 0.126 in model (4) and 0.3 in model (5). The positive elasticity of capital is the result of a large investment in capital infrastructure and fixed capital formation. Over the years, government-related expenditure towards capital investment has increased which represents on average 30 percent of the total government expenditure [36]. In the short-run, capital per capita $\Delta lnk_t$ in model (5) is negative and statistically insignificant.

The education variable, secondary school enrolment rate ($SEC_t$) has the expected positive sign while being statistically significant at the 10 percent level in each model. The results imply

**Table 5. Long-run results of model (4), 1972–2016.**

| Variables (dependent is *lny*) | Coefficients | t-Statistic |
|--------|------|------|
| *Constant* | 6.586[A] | 23.216 |
| *lnk* | 0.126 [A] | 2.972 |
| *SEC* | 0.0022 [A] | 6.417 |
| *NCD* | -0.012 [A] | -2.840 |

A, B, C indicate significance at 1, 5 and 10 percent levels.

**Table 6. Short-run results of model (4), 1972–2016.**

| Variables (dependent is $\Delta lny$) | Coefficients | t-Statistic |
|---|---|---|
| $\Delta lny_{t-1}$ | 0.303$^C$ | 1.812 |
| $\Delta lny_{t-2}$ | 0.451 $^A$ | 3.400 |
| $\Delta lny_{t-3}$ | 0.284$^B$ | 2.207 |
| $\Delta SEC_t$ | 0.006$^B$ | 2.067 |
| $\Delta SEC_{t-1}$ | -0.008 $^A$ | -3.146 |
| $\Delta NCD_t$ | -0.005 $^A$ | -2.934 |
| $\Delta NCD_{t-1}$ | 0.005 $^A$ | 3.483 |
| $\Delta NCD_{t-2}$ | 0.006 $^A$ | 4.230 |
| COUP | -0.075 $^A$ | -3.228 |
| $ECT_{t-1}$ | -0.698 $^A$ | -4.417 |
| **Goodness of fit and diagnostic tests** | | |
| Observations | | 45 |
| $\chi^2(sc)$– Serial Correlation | | 0.705[0.504] |
| $\chi^2(hs)$—Heteroscedasticity | | 0.386[0.963] |
| $\chi^2(ff)$–Functional Form | | 0.864[0.361] |
| $\chi^2(n)$–Normality | | 1.598[0.450] |

A, B, C indicate significance at 1, 5 and 10 percent levels. F-statistics represent diagnostic tests for serial correlation, heteroscedasticity and functional form while Jarque-Bera test statistic is shown for normality test. [] contains the p-values.

that development of human capital is crucial for Fiji to increase the output per capita. With this in mind, the Government of Fiji has provided various scholarships and loans schemes to the students to develop the human capital capacity. The Government of Fiji has also made primary and secondary education free. Our results suggest that the current initiative by the Government of Fiji in terms of investment in education will significantly improve the output of Fiji in the future. In the short-run, $\Delta SEC_t$ has a positive sign and is statistically significant at the 10 percent level in both models. Meanwhile, $\Delta SEC_{t-1}$ has a negative sign, while being statistically significant in both models. The negative coefficient implies that the benefits of education are not realised in the short-run since it takes many years to develop the human capital capacity.

In both models, the dummy variable (COUP) for political instabilities in years 1987, 2000 and 2006 had a negative sign and is strongly significant at 1 percent level in the short-run. The

**Table 7. Long-run results of model (5), 1972–2016.**

| Variables (dependent is $lny$) | Coefficients | t-Statistic |
|---|---|---|
| Constant | 6.555$^A$ | 13.029 |
| lnk | 0.300 $^A$ | 4.404 |
| SEC | 0.0081 $^C$ | 1.887 |
| DM | -0.010 $^A$ | -3.003 |
| CVD | -0.018 $^A$ | -3.665 |
| CRD | -0.031 $^C$ | -1.911 |
| CAN | -0.035 $^A$ | -4.212 |

A, B, C indicate significance at 1, 5 and 10 percent levels.

**Table 8. Short-run results of model (5), 1972–2016.**

| Variables (dependent is $\Delta lny$) | Coefficients | t-Statistic |
|---|---|---|
| $\Delta lny_{t-1}$ | 0.204 [C] | 1.842 |
| $\Delta lny_{t-2}$ | 0.274 [B] | 2.475 |
| $\Delta lny_{t-3}$ | 0.343 [A] | 2.991 |
| $\Delta lnk_t$ | -0.003 | -0.051 |
| $\Delta SEC_t$ | 0.004 [C] | 1.985 |
| $\Delta SEC_{t-1}$ | -0.007 [A] | -3.371 |
| $\Delta SEC_{t-2}$ | 0.003 [C] | 2.063 |
| $\Delta CVD_t$ | -0.009 [A] | -5.313 |
| $\Delta CVD_{t-1}$ | 0.005 [A] | 3.740 |
| $\Delta CVD_{t-2}$ | 0.005 [A] | 4.263 |
| $\Delta CRD$ | -0.014 [A] | -3.802 |
| $\Delta CRD_{t-1}$ | 0.012 [A] | 3.284 |
| $\Delta CAN$ | -0.0023 | -0.882 |
| $\Delta CAN_{t-1}$ | -0.020 [A] | -5.976 |
| $\Delta CAN_{t-2}$ | 0.011 [A] | 4.340 |
| $COUP$ | -0.084 [A] | -4.416 |
| $\Delta ECT_{t-1}$ | -0.777 [A] | -7.305 |
| **Goodness of fit and diagnostic tests** | | |
| Observations | | 45 |
| $R^2$ | | 0.808 |
| $\bar{R}^2$ | | 0.680 |
| $\chi^2(sc)$–Serial Correlation | | 1.052[0.374] |
| $\chi^2(hs)$—Heteroscedasticity | | 0.480[0.949] |
| $\chi^2(ff)$–Functional Form | | 0.534[0.476] |
| $\chi^2(n)$–Normality | | 1.801[0.406] |

A, B, C indicate significance at 1, 5 and 10 percent levels. F-statistics represent diagnostic tests for serial correlation, heteroscedasticity and functional form while Jarque-Bera test statistic is shown for normality test. [] contains the p-values.

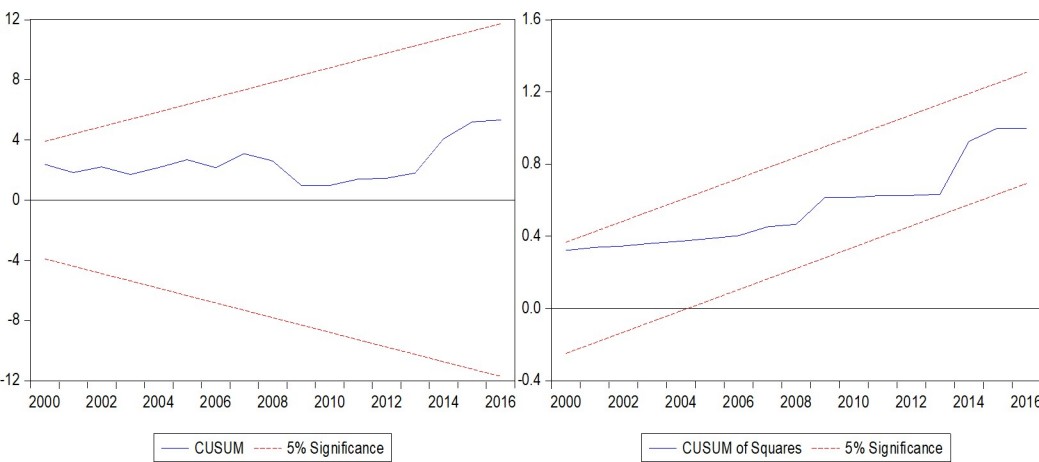

**Fig 1. Model (4) CUSUM and CUSUM of squares test.** CUSUM and CUSUM of Squares are tests for model stability. At 5 percent significance level, the lines stay within the critical bounds, indicating long-run model stability.

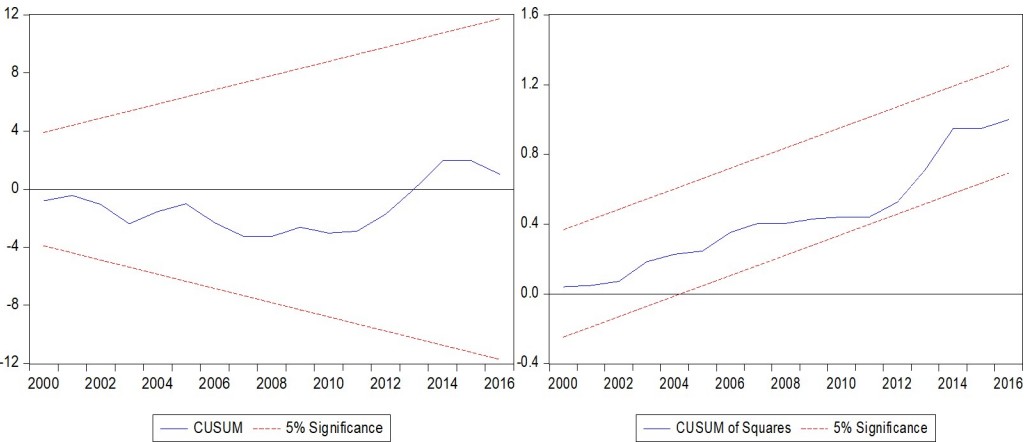

**Fig 2. Model (5) CUSUM and CUSUM of squares test.** CUSUM and CUSUM of Squares are tests for model stability. At 5 percent significance level, the lines stay within the critical bounds, indicating long-run model stability.

results indicate that coups negatively affect the output per capita of Fiji, which is consistent with the findings of Narayan and Smyth [35] and Chand [37].

Furthermore, the non-communicable disease mortality rate ($NCD_t$) has the expected negative sign in the long-run and is statistically significant at 1 percent level in model (4). The negative sign implies that NCD-related mortality negatively affects the long-run output per capita since it causes death of workers in the working-age population of Fiji, which reduces the size of the skilled and unskilled labour force. In Fiji, NCDs represent more than 80 percent of the total mortality, of which most deaths are premature, that is, before the retirement age. A high rate of NCD-related deaths among the working age population also increases the years of productive life lost, which negatively affects the output per capita.

In the short-run, we found that $\Delta NCD_t$ has a negative sign in model (4) and is statistically significant at 5 percent level. Meanwhile, $\Delta NCD_{t-1}$ and $\Delta NCD_{t-2}$ have a positive sign and are statistically significant at 5 percent level. Hence, the results imply that immediate NCD mortality reduces the output due to loss of skilled workers. However, past years' NCD mortality rate has a positive sign, which indicates that firms are able to employ new workers to fill in vacant positions left by deceased workers and restore production in one to two years following the death of a worker.

Furthermore, we decomposed the aggregate NCD mortality rate into four major types of NCDs to measure the individual impact on output per capita. Diabetes ($DM$), cardiovascular diseases ($CVD$), chronic respiratory disease ($CRD$), and cancer ($CAN$) mortality rates have the expected negative sign in the long-run in model (5) (Table 7). The mortality rates of four major types of NCDs are also statistically significant at 10 percent significance level. Hence, we found that the aggregate NCD mortality rates, as well as the four major types of NCDs, have a deteriorating impact on the long-run output per capita of Fiji. We also found that in the short-run, $\Delta CVD_t$, $\Delta CRD_t$ and $\Delta CAN_{t-1}$ have a significant negative effect on the output per capita. However, $\Delta CVD_{t-1}$, $\Delta CVD_{t-2}$, $\Delta CRD_{t-1}$, and $\Delta CAN_{t-2}$ have a significant positive effect on output, which suggests that firms are able to replace workers who died due to cardiovascular disease, chronic respiratory disease and cancer in one to two years following the death of a worker and able to restore the level of output.

The overall result of this study is in line with the theory of Suhrcke and Urban [19] and findings of Frank [23] where they argued that as NCD-related mortality become more common in the developing countries, it will start to negatively affect the long-run output per capita.

## Robustness test

We re-estimated model (5) with additional control variables to check the robustness and consistency of the coefficients with the results presented in Table 9 (long-run) and Table 10 (short-run). Some of the important determinants of output in Fiji are trade openness [38], government expenditure [36], and inflation rate [39]. Hence, we added each control variable one at a time and estimated the model using the ARDL framework. Model (i) includes trade openness, which controls the supply and availability of healthy and unhealthy food products in the economy [40]. Trade openness captures bad nutrition habits that lead to the development of NCDs in the population. Furthermore, the government expenditure variable captures the role of government in providing funds for NCD-related preventative and curative services for the formulation of NCD strategic plans and free medicine schemes [41]. Lastly, the inflation rate captures the change in the price of nutritious and healthy food. We assume that as the price of nutritious and healthy foods increase, individuals will buy inexpensive processed foods as alternatives that potentially increase the risk of NCDs in the population [42].

Overall, we find that the four major types of NCD mortality rates have the expected negative sign and are significant at the 10 percent level. The coefficients of diabetes, cardiovascular disease, chronic respiratory disease and cancer mortality rates are significant and moderately higher than the long-run estimates of model (5).

The inclusion of trade openness *(OPENt)*, and government expenditure *(GOVt)* as additional variables in models (i) and (ii) show very little fluctuations in the coefficients of diabetes, cardiovascular disease, chronic respiratory disease and cancer mortality rates. The only difference being that *SEC* became weakly insignificant in model (i). The largest fluctuation in the coefficient value is noticed when the inflation rate *(INFt)* was added as an additional variable in model (iii). However, the coefficients of trade openness, government expenditure, and inflation rate were highly insignificant despite having the expected sign.

Hence, the robustness test shows the impact range of four major types of NCD mortality rates on the level of output per capita. On average, the coefficient of diabetes (DM) ranges from -0.009 to -0.019, cardiovascular disease (CVD) ranges from -0.015 to -0.031, chronic respiratory disease (CRD) ranges from -0.026 to -0.037 and cancer (CAN) ranges from -0.033 to -0.042. Hence, diabetes, cardiovascular disease, chronic respiratory disease and cancer mortality lowered Fiji's output per capita from 1972 to 2016 and the effects will be heavier in future when the number of NCD deaths rise.

**Table 9. Effect of NCD mortality on output per capita–robustness test.**

| Variables ($lny_t$) | Models | | |
|---|---|---|---|
| | (i) | (ii) | (iii) |
| *Added control variable* | *Trade openness* | *Government Expenditure* | *Inflation Rate* |
| constant | 6.103 [A] (10.661) | 6.734 [A] (14.406) | 4.321 [A] (404) |
| $lnk_t$ | 0.289 [A] (4.399) | 0.263 [A] (4.208) | 0.511 [A] (4.033) |
| $DM_t$ | -0.011 [A] (-3.134) | -0.009 [A] (-3.302) | -0.019 [A] (-3.548) |
| $CVD_t$ | -0.022 [A] (-3.058) | -0.015 [A] (-3.349) | -0.031 [A] (-3.932) |
| $CRD_t$ | -0.037 [C] (-1.944) | -0.030 [B] (-2.113) | -0.026 [C] (-1.784) |
| $CAN_t$ | -0.038 [A] (-4.012) | -0.036 [A] (-4.811) | -0.042 [A] (-5.056) |
| $SEC_t$ | 0.0070 (1.502) | 0.0079 [C] (2.065) | 0.0090 [C] (2.007) |
| $OPEN_t$ | 0.001 (1.018) | - | - |
| $GOV_t$ | - | 0.006 (1.241) | - |
| $INF_t$ | - | - | -0.005 (-1.627) |

A, B, C indicate significance at 1, 5 and 10 percent levels.

Table 10. Error correction representations for the selected ARDL model–robustness test.

| Variables ($\Delta lny_t$) | Models | | |
|---|---|---|---|
| | (i) | (iii) | (iv) |
| *Added control variable* | *Trade openness* | *Government Expenditure* | *Inflation Rate* |
| $\Delta lny_{t-1}$ | 0.160 (1.506) | 0.206 [C] (1.902) | 0.330 [A] (2.911) |
| $\Delta lny_{t-2}$ | 0.153 (1.337) | 0.262 [B] (2.458) | - |
| $\Delta lny_{t-3}$ | 0.300 [A] (2.662) | 0.350 [A] (3.168) | - |
| $\Delta lnk_t$ | -0.046 (-0.920) | -0.003 (-0.052) | 0.008 (0.175) |
| $\Delta lnk_{t-1}$ | - | - | -0.110 [C] (-1.809) |
| $\Delta lnk_{t-2}$ | - | - | -0.281 [A] (-3.999) |
| $\Delta DM_t$ | - | - | -0.010 [A] (-4.790) |
| $\Delta CVD_t$ | -0.010 [A] (-5.938) | -0.009 [A] (-556) | -0.012 [A] (-6.989) |
| $\Delta CVD_{t-1}$ | 0.007 [A] (4.753) | 0.004 [A] (3.354) | 0.009 [A] (576) |
| $\Delta CVD_{t-2}$ | 0.006 [A] (5.045) | 0.004 [A] (3.716) | 0.007 [A] (6.260) |
| $\Delta CRD_t$ | -0.015 [A] (-4.308) | -0.014 [A] (4.050) | -0.007 [C] (-2.019) |
| $\Delta CRD_{t-1}$ | 0.016 [A] (4.166) | 0.012 [A] (3.548) | 0.013 [A] (3.962) |
| $\Delta CAN_t$ | -0.002 (-0.835) | -0.003 (-1.143) | 0.0002 (0.076) |
| $\Delta CAN_{t-1}$ | 0.021 [A] (6.285) | 0.022 [A] (6.546) | 0.022 [A] (6.307) |
| $\Delta CAN_{t-2}$ | 0.010 [A] (4.299) | 0.012 [A] (4.868) | 0.011 [A] (4.454) |
| $\Delta SEC_t$ | 0.003 (1.596) | 0.004 [C] (2.046) | 0.008 [A] (4.525) |
| $\Delta SEC_{t-1}$ | -0.007 [A] (-3.573) | -0.007 [A] (-3.800) | -0.003 [C] (-2.018) |
| $\Delta SEC_{t-2}$ | 0.005 [A] (2.826) | 0.003 [C] (1.725) | 0.006 [A] (3.958) |
| $\Delta OPEN_t$ | -0.0004 (-0.925) | - | - |
| COUP | -0.075 [A] (-4.141) | -0.085 [A] (-4.638) | -0.110 [A] (-5.308) |
| $ECT_{t-1}$ | -0.805 [A] (-7.704) | -0.874 [A] (-7.729) | -0.687 [A] (-7.727) |
| **Goodness of Fit and diagnostic tests** | | | |
| $\bar{R}^2$ | 0.704 | 0.704 | 0.716 |
| Standard Error | 0.023 | 0.023 | 0.022 |
| $\chi^2(sc)$–Serial Correlation | 0.636 [0.545] | 0.876 [0.438] | 0.071 [0.932] |
| $\chi^2(hs)$—Heteroscedasticity | 2.418 [0.033] | 0.547 [0.912] | 0.521 [0.930] |
| $\chi^2(ff)$–Functional Form | 0.156 [0.699] | 2.642 [0.125] | 0.604 [0.449] |
| $\chi^2(n)$—Normality | 1.064 [0.588] | 1.918 [0.383] | 2.024 [0.364] |
| Bounds test | 4.301 [B] | 4.425 [B] | 4.422 [A] |

A, B, C indicate significance at 1, 5 and 10 percent levels. F-statistics represent diagnostic tests for serial correlation, heteroscedasticity and functional form while Jarque-Bera test statistic is shown for normality test. [] contains the p-values.

The coefficients of the error correction terms were -0.81, -0.87 and -0.69 for models (i), (ii) and (iii) respectively, which suggests that convergence to equilibrium was rapid. The Adjusted-R squared also improved by an average of 2.4 to 3.6 percent. In addition, the models passed the diagnostic tests for serial correlation, heteroscedasticity, normality and functional form, except for model (i) that failed to pass the test for heteroscedasticity. However, it is common to find the problem of heteroscedasticity in the ARDL framework since the models use variables that are both integrated of order zero and order one [33].

## Conclusion and policy implications

The ARDL bounds test confirmed that there exists a long-run cointegration relationship among the variables when output per capita is the dependent variable, while capital per

worker, secondary schooling and mortality rates for diabetes, cardiovascular disease, chronic respiratory disease, cancer and NCDs are the independent in models (4) and (5). In model (4), capital per worker and secondary schooling have a positive sign while NCD mortality rate has the expected significant negative sign. In model (5), diabetes, cardiovascular disease, chronic respiratory disease and cancer mortality rates had a significant negative effect on the long-run output per capita. The robustness test also confirmed that the parameters remain consistent in terms of signs across different scenarios. Afterwards, the error correction model was estimated to derive the short-run relationship. The negative and highly significant error correction term also confirmed a long-run relationship among the variables and the convergence towards the steady state was quick.

The empirical studies on the relationship between NCD mortality rate and output per capita have shown mixed results. According to Suhrcke and Urban [19], CVD mortality rate negatively affects the output of developed countries, while there exists a positive relationship between CVD mortality rate and output in developing countries. Our results contradict Suhrcke and Urban [19] since we found that NCD mortality rates have a significant negative effect on the output of Fiji, which is a developing country. On a similar note, Frank [23] also found that NCD mortality negatively affects output in Latin America and the Caribbean countries. Hence, based on the results of this study and Frank (2014), we conclude that NCD mortality rates have begun to negatively affect the output per capita of developing countries.

The study has several limitations; hence, the results of this should be used and interpreted cautiously. Firstly, the study uses the NCD mortality rate as a proxy to measure the impact of the non-communicable diseases on Fiji's output per capita due to the lack of data on NCD morbidity, disability, prevalence and incidence rate in Fiji. Secondly, we were not able to test the robustness of the results using other cointegration techniques apart from the ARDL bounds test approach since the variables of interest were integrated of both order zero and one. Thirdly, the study measures the impact of NCD mortality rate on the long-run output per capita of Fiji and not the economic growth. Factor accumulation, human capital and NCD mortality rates were only able to explain 70 to 80 percent of the variation in output per capita, hence, to get an in-depth view on the impact of NCD mortality on output per capita of Fiji, additional explanatory variables will need to be augmented in the model.

Overall, our finding have important policy implication for Fiji. Policy makers should make serious effort to lower NCD related fertility rate in Fiji. This necessitates measures such as multi-sectoral collaboration and collective response from various organisations such as households, businesses, NGOs and the government. Individuals can prevent the indirect cost of NCD-related mortality by changing lifestyle behaviour. A healthy diet, taking part in physical activities as well as avoiding the consumption of alcohol and tobacco are best remedies that individuals can practise to avoid the risk of NCDs.

Similarly, the Fijian Government should extensively fund the NCD preventative agenda instead of just in NCD curative areas. According to a policy brief from Ministry of Health and Medical Services [43], Government allocated fund are usually larger for curative purposes compared to preventative purposes. The NCD preventative policy should focus on providing education on lifestyle diseases, impose subsidy on healthy foods and tax the harmful foods, tobacco and alcohol. Physical activity should be highly encouraged among adults and ensure compulsory participation in physical activities in schools. These initiatives are necessary on households and government's part to reduce the risk of contracting NCDs and its negative implications on the economy.

Businesses should maintain a certain requirement of healthy living and engagement in physical activity among the employees since it increase business expenditure for employee healthcare and coverage at the firm level [44]. Although it requires some investment, in the

long-run healthy and skilled workers will improve the level of output and productivity. This will also reduce the cost of training and developing workers to fill the gap left by deceased or absent worker suffering from NCDs.

## Supporting information

**S1 Data.**
(XLSX)

## Author Contributions

**Conceptualization:** Shamal Shivneel Chand, Baljeet Singh.

**Data curation:** Shamal Shivneel Chand, Baljeet Singh.

**Formal analysis:** Shamal Shivneel Chand, Baljeet Singh, Sanjesh Kumar.

**Funding acquisition:** Shamal Shivneel Chand.

**Investigation:** Shamal Shivneel Chand, Baljeet Singh, Sanjesh Kumar.

**Methodology:** Shamal Shivneel Chand, Baljeet Singh.

**Resources:** Shamal Shivneel Chand, Baljeet Singh.

**Software:** Shamal Shivneel Chand, Baljeet Singh, Sanjesh Kumar.

**Supervision:** Baljeet Singh, Sanjesh Kumar.

**Visualization:** Baljeet Singh.

**Writing – original draft:** Shamal Shivneel Chand, Baljeet Singh.

**Writing – review & editing:** Shamal Shivneel Chand, Baljeet Singh, Sanjesh Kumar.

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
