## [Decision Letter · Decision Letter 0]

20 May 2020

PONE-D-20-10474

Non-Communicable disease mortality in Small Pacific Island Countries: Estimating the loss of human capital resource

PLOS ONE

Dear Dr. Singh,

Thank you for submitting your manuscript to PLOS ONE. After careful consideration, we feel that it has merit but does not fully meet PLOS ONE’s publication criteria as it currently stands. Therefore, we invite you to submit a revised version of the manuscript that addresses the points raised during the review process.

We would appreciate receiving your revised manuscript by Jul 04 2020 11:59PM. To enhance the reproducibility of your results, we recommend that if applicable you deposit your laboratory protocols in protocols.io, where a protocol can be assigned its own identifier (DOI) such that it can be cited independently in the future. For instructions see: http://journals.plos.org/plosone/s/submission-guidelines#loc-laboratory-protocols

We look forward to receiving your revised manuscript.

Kind regards,

Gausal A Khan, Ph.D

Academic Editor

PLOS ONE

Journal Requirements:

2. Please ensure you have thoroughly discussed any potential limitations of this study within the Discussion section.

Additional Editor Comments (if provided):

Dear Author.

Please go though the reviewers comments and point-to-point reply will submit for further evaluation.

Reviewers' comments:

Reviewer's Responses to Questions

**Comments to the Author**

1. Is the manuscript technically sound, and do the data support the conclusions?

Reviewer #1: Partly

Reviewer #2: Yes

2. Has the statistical analysis been performed appropriately and rigorously? 

Reviewer #1: N/A

Reviewer #2: No

3. Have the authors made all data underlying the findings in their manuscript fully available?

Reviewer #1: No

Reviewer #2: Yes

4. Is the manuscript presented in an intelligible fashion and written in standard English?

Reviewer #1: Yes

Reviewer #2: Yes

5. Review Comments to the Author

Reviewer #1: Results have to organised in a better way and figures should be more presentable. Discussion should be brief and to the point. Conclusion does not reflect discussion, it should be rewritten. Some of the references are not organised properly.

Reviewer #2: Health economics is a growing field that push the universal health coverage agenda forward for improving health coverage, expanding services, and providing financial-risk protection are all necessary to optimize the population health. Chronic non-communicable disease (NCD) is the big issue in low and middle income countries which really need to focus on the building health systems that integrate complex, multidisciplinary interventions to reduce the economical burden.

This study is focused on estimate the net percentage value loss of non-health GDP, and indirect cost of Non-communicable disease related mortality among the working age group of eight small Pacific Island countries (PIC) in 2017.Using human capital approach methodology with non-health GDP per capita, author emphasize that NCD related premature death in working age population creates a potential economical burden for the PICs.

In most of the countries, the premature mortality due to cardiovascular disease and in working ages from 15 until the retirement age. Also, this study has recommended intervention to reduce the prevalence of NCD related that all small PICs should reduce the prevalence of NCD related premature death.

This manuscript well written and describe the economical burden for the small Pacific Islands countries and also the interventions to reduce.

The following comments should be resolved:

(1) Which statistical analysis was used in this study to claim that NCD-related death in the working age group create a significant economic burden in the PIC?

(2) Figure 1, bar diagram shows that percentage of total death is increased in NCD group in different group, but it is not clear that it is significant or not as no statistical analysis was shown.

(3) Same in case of figure 3.

6. PLOS authors have the option to publish the peer review history of their article (what does this mean?). If published, this will include your full peer review and any attached files.

Reviewer #1: No

Reviewer #2: No

---

## [Author Response · Author response to Decision Letter 0]

18 Jun 2020

Response to Reviewers

Comments to the Author

1. Is the manuscript technically sound, and do the data support the conclusions?

Reviewer #1: Partly

Reviewer #2: Yes

Response: We have revised the methodology and the methods used that are technically sound. We have used the Autoregressive Distributed Lag (ARDL) bounds test to cointegration approach to draw our conclusion. We have used official secondary data which is compiled by Ministry of Health, Fiji and the World Health Organization. Sample used is appropriate and bounds test to cointegration is standard procedure used to analyse time series data. Revised manuscript is based on Fiji as there no data available for other Pacific Island countries. Revised manuscript is based on long time series data which is only available for Fiji. 

2. Has the statistical analysis been performed appropriately and rigorously?

Reviewer #1: N/A

Reviewer #2: No

Response: We have revised the methodology. We have used bounds test to cointegration approach. One of the advantages of the ARDL model is that it is very flexible where variables of both I(0) and I(1) status can be used to test for a cointegrating relationship. Secondly, the ARDL approach is an advantage to studies using a small sample size for estimation and forecasting [29]. Thirdly, the ARDL approach gives a consistent and unbiased estimate of the long-run parameters and each variable can have their own different lag-lengths compared to conventional cointegration tests. The fourth advantage of the ARDL approach is that it adequately deals with the problems of autocorrelation and endogeneity and provides unbiased and super-consistent coefficients with valid t-statistics. In sum, there are two steps involved in the ARDL bounds test approach to cointegration. We performed a number of robustness test and found our result is consistent. We carried out several diagnostic test and our result suffice all classical assumption.

3. Have the authors made all data underlying the findings in their manuscript fully available?

Reviewer #1: No

Reviewer #2: Yes

Response: We have uploaded all the data used in the analysis.

4. Is the manuscript presented in an intelligible fashion and written in standard English?

Reviewer #1: Yes

Reviewer #2: Yes

Response: Manuscript presented and written in standard English.

5. Review Comments to the Author

Reviewer #1: Results have to organised in a better way and figures should be more presentable. Discussion should be brief and to the point. Conclusion does not reflect discussion, it should be rewritten. Some of the references are not organised properly.

Response: We have rewritten all our results and discussion. Now it is more presentable, clear and easy to follow. We have rewritten our conclusion and policy implications and now it reflects the findings and discussion. 

Reviewer #2: Health economics is a growing field that push the universal health coverage agenda forward for improving health coverage, expanding services, and providing financial-risk protection are all necessary to optimize the population health. Chronic non-communicable disease (NCD) is the big issue in low and middle income countries which really need to focus on the building health systems that integrate complex, multidisciplinary interventions to reduce the economical burden.

This study is focused on estimate the net percentage value loss of non-health GDP, and indirect cost of Non-communicable disease related mortality among the working age group of eight small Pacific Island countries (PIC) in 2017.Using human capital approach methodology with non-health GDP per capita, author emphasize that NCD related premature death in working age population creates a potential economical burden for the PICs.

In most of the countries, the premature mortality due to cardiovascular disease and in working ages from 15 until the retirement age. Also, this study has recommended intervention to reduce the prevalence of NCD related that all small PICs should reduce the prevalence of NCD related premature death.

This manuscript well written and describe the economical burden for the small Pacific Islands countries and also the interventions to reduce.

The following comments should be resolved:

(1) Which statistical analysis was used in this study to claim that NCD-related death in the working age group create a significant economic burden in the PIC?

Response: Our revised manuscript is extensively rewritten based on the comments and suggestion of two reviewers. In the revised manuscript, we have used bounds test to cointegration approach, it is standard procedure used to analyse factors affecting economic output. We carried out a several robustness test and find out result is consistent. Revised manuscript is based on Fiji as there no data available for other Pacific Island countries. Revised manuscript is based on long time series data which is only available for Fiji.

(2) Figure 1, bar diagram shows that percentage of total death is increased in NCD group in different group, but it is not clear that it is significant or not as no statistical analysis was shown.

Response: All results and graph figures are revised based on our revised findings. We have used ARDL bounds test to analyse our data. We carried out a number of diagnostic test and find our result is reliable.

(3) Same in case of figure 3.

Response: All our analysis is based on ARDL bounds testing technique. However, this comment is no longer relevant in revised manuscript.

---

## [Decision Letter · Decision Letter 1]

29 Jun 2020

The economic burden of non-communicable disease mortality in the South Pacific: Evidence from Fiji

PONE-D-20-10474R1

Dear Dr. Singh,

We’re pleased to inform you that your manuscript has been judged scientifically suitable for publication and will be formally accepted for publication once it meets all outstanding technical requirements.

Kind regards,

Gausal A Khan, Ph.D

Academic Editor

PLOS ONE

Additional Editor Comments (optional):

Reviewers' comments:

Reviewer's Responses to Questions

**Comments to the Author**

1. If the authors have adequately addressed your comments raised in a previous round of review and you feel that this manuscript is now acceptable for publication, you may indicate that here to bypass the “Comments to the Author” section, enter your conflict of interest statement in the “Confidential to Editor” section, and submit your "Accept" recommendation.

Reviewer #1: All comments have been addressed

Reviewer #2: All comments have been addressed

2. Is the manuscript technically sound, and do the data support the conclusions?

Reviewer #1: Yes

Reviewer #2: Yes

3. Has the statistical analysis been performed appropriately and rigorously? 

Reviewer #1: Yes

Reviewer #2: Yes

4. Have the authors made all data underlying the findings in their manuscript fully available?

Reviewer #1: Yes

Reviewer #2: Yes

5. Is the manuscript presented in an intelligible fashion and written in standard English?

Reviewer #1: Yes

Reviewer #2: Yes

6. Review Comments to the Author

Reviewer #1: The authors responded to the comments made by the reviewer. The results should be described in the simplified manner. Reference need to be corrected.

Reviewer #2: This study is interesting and valuable regarding the economic burden and non-communicable disease mortality in the south Pacific, especially Fiji. The revised manuscript described well about the methods and statistical analysis based on comments and explains the analysis of the data. I am fully recommending this revised manuscript for publication. I hope, this publication opens a new sight to think about the economic burden and non-communicable related mortality rate in other countries.

7. PLOS authors have the option to publish the peer review history of their article (what does this mean?). If published, this will include your full peer review and any attached files.

Reviewer #1: No

Reviewer #2: No

---

## [Editor Report · Acceptance letter]

6 Jul 2020

PONE-D-20-10474R1 

The economic burden of non-communicable disease mortality in the South Pacific: Evidence from Fiji 

Dear Dr. Singh:

I'm pleased to inform you that your manuscript has been deemed suitable for publication in PLOS ONE. Congratulations! Your manuscript is now with our production department. 

Kind regards, 

on behalf of

Dr. Gausal A Khan 

Academic Editor

PLOS ONE